# Is Autophagy Inhibition in Combination with Temozolomide a Therapeutically Viable Strategy?

**DOI:** 10.3390/cells12040535

**Published:** 2023-02-07

**Authors:** Ahmed M. Elshazly, David A. Gewirtz

**Affiliations:** 1Department of Pharmacology and Toxicology, Massey Cancer Center, Virginia Commonwealth University, 401 College St., Richmond, VA 23298, USA; 2Department of Pharmacology and Toxicology, Faculty of Pharmacy, Kafrelsheikh University, Kafrelsheikh 33516, Egypt

**Keywords:** temozolomide, autophagy, melanoma, glioblastoma, cytoprotective, cytotoxic

## Abstract

Temozolomide is an oral alkylating agent that is used as the first line treatment for glioblastoma multiform, and in recurrent anaplastic astrocytoma, as well as having demonstrable activity in patients with metastatic melanoma. However, as the case with other chemotherapeutic agents, the development of resistance often limits the therapeutic benefit of temozolomide, particularly in the case of glioblastoma. A number of resistance mechanisms have been proposed including the development of cytoprotective autophagy. Cytoprotective autophagy is a survival mechanism that confers upon tumor cells the ability to survive in a nutrient deficient environment as well as under external stresses, such as cancer chemotherapeutic drugs and radiation, in part through the suppression of apoptotic cell death. In this review/commentary, we explore the available literature and provide an overview of the evidence for the promotion of protective autophagy in response to temozolomide, highlighting the possibility of targeting autophagy as an adjuvant therapy to potentially increase the effectiveness of temozolomide and to overcome the development of resistance.

## 1. Introduction

This manuscript is one of a series of papers that explore the role of autophagy in response to different cancer therapeutic modalities. Our previous publications covered radiation [1], cisplatin [2], microtubule poisons [3], hormonal therapies in estrogen positive breast cancer [4], and, most recently, PARP inhibitors [5], and Topoisomerase I poisons [6]. Our overarching goal is to determine whether there are particular therapeutic modalities where the preclinical data, and where available, clinical trials, support the inclusion of autophagy inhibition as an adjuvant approach.

## 2. Autophagy Overview

Autophagy is a self-degradative process whereby the cell recycles damaged proteins and organelles to maintain cellular hemostasis [3,7,8]. Autophagy tends to occur at a basal level in all cells and can further be triggered by conditions of nutrient deprivation and starvation, serving as a source of energy production [3,9,10]. Autophagy can also be induced in response to other forms of cellular stress, including hypoxia, oxidative stress, endoplasmic reticulum (ER) stress, and protein aggregation [3,11,12]. Autophagy is a multistep process that is regulated by several highly conserved autophagy (ATG) proteins, and is frequently initiated via the activation of different signaling pathways, specifically a decrease in mammalian target of rapamycin (mTOR) activity and an increase of Unc-51 like autophagy activating kinase 1 (ULK1) activity. ULK1 is consequently dissociated from the 5′ adenosine monophosphate-activated protein kinase (AMPK), leading to autophagy activation [4,13,14,15]. Autophagy can also be trigged under hypoxic conditions through the hypoxia-inducible factor 1-α (HIF-α). HIF-1α accumulation activates BNIP/BNIP3L expression, which subsequently dissociates the complex between Bcl-2 and Beclin-1 to initiate autophagy [13,16]. Functionally, autophagic flux begins with formation of the phagophore, a double membrane structure that encompasses damaged cytoplasmic constituents [17]; subsequently, phagophores extend with cytoplasm engulfment, forming autophagosomes, which then fuse with lysosomes, resulting in autolysosome formation for the degradation of cytoplasmic cargo [3,17]. For more details, the mechanism of autophagy is discussed by Rubinsztein et al. [18] and in our previous publications [3,4].

Many chemotherapeutic agents have been shown to induce autophagy in different types of malignancies. Multiple functional forms of autophagy have been identified, which we have classified into four groups, specifically cytotoxic, cytoprotective, and non-protective, as well as cytostatic forms [19]. The major form that has been studied by many laboratories as a potential therapeutic strategy for increasing the effectiveness of different classes of chemotherapeutic drugs as well as overcoming resistance is cytoprotective autophagy. Cytoprotective autophagy acts as a survival mechanism that confers upon tumor cells the ability to be shielded from starvation and to evade apoptotic signals [20]. There are many reports in the literature where the activation of cytoprotective autophagy, which reduces the sensitivity of tumor cells to chemotherapeutic drugs and radiation, is associated with drug resistance [21]. Therefore, targeting the cytoprotective form of autophagy is considered as a potential therapeutic modality in the treatment of different types of malignancies, generally by utilizing clinically approved autophagy inhibitors, such as hydroxychloroquine (HCQ) [3]. Recently, a number of publications studied the possible modulation of the dysregulated autophagic process. Sharma et al. [22] discussed the possible utilization of miRNAs, especially miRNA-34a, to modulate autophagy. Furthermore, Rahman et al. [23] covered the possible modulation of p53 in autophagy signaling pathways.

## 3. Temozolomide

Alkylating agents are one of the oldest classes of antitumor drugs that are currently used in the clinical setting for treatment of various hematologic and solid malignancies. Among these, Temozolomide (TMZ) is an oral alkylating agent that is the first line of treatment for glioblastoma multiform (GMB). GBM is the most common and aggressive malignant brain tumor in adults, accounting for approximately 14.5% of all central nervous system tumors and 48.6% of malignant central nervous system tumors, with a median survival of 15 months [24,25]. Up to 70% of GBM patients will experience disease progression within one year of diagnosis [26], with fewer than 5% of patients surviving five years after diagnosis [27]. 

The current standard-of-care for glioblastoma includes surgery and radiation in combination with temozolomide. Unfortunately, patient prognosis remains poor, with the majority of glioblastoma patients exhibiting disease relapse. Once tumors progress after first-line therapy, treatment options are limited and management of recurrent glioblastoma remains challenging [28] with the absence of effective therapeutic strategies or the development of drug resistance. TMZ is also used in treatment of recurrent anaplastic astrocytoma [29] and also has demonstrable activity in patients with metastatic melanoma [30].

After its absorption, TMZ undergoes intracellular conversion via hydrolysis into a potent methylating agent, monomethyl triazeno imidazole carboxamide (MTIC). TMZ-mediated cytotoxicity results primarily from the formation of DNA-methyl adducts, preferentially at the O^6^ position of guanine [29,31]. MTIC alkylation also occurs at both the N^7^ position of guanine and the N^3^ position of adenine [31]. N^7^ and N^3^ alkylation are susceptible to base excision repair (BER). The alkylation at the N^7^ position of guanine appears not to be markedly cytotoxic, whereas alkylation at the N^3^ position of adenine tends to be lethal if not intercepted [32]. With regard to O^6^-guanine base methylation, the enzyme methylguanine-DNA methyltransferase (MGMT) can eliminate the methyl adduct from the O^6^ position of guanine. However, when the methylation is unrepaired, the guanine bases can mispair with thymine (rather than the natural partnering base, cytosine) during DNA replication, activating DNA mismatch repair (MMR) [32,33,34]. MMR can recognize the mispaired bases in the DNA daughter strand, with their subsequent excision; however, O^6^-guanine bases persist in the template strand. Therefore, ineffective cycles of thymine reinsertion and excision result in persistent DNA strand breaks and subsequent cell death (Figure 1) [35]. Consequently, functional MMR and low levels of MGMT are required for the therapeutic effectiveness of TMZ [32].

As is frequently the case with most cancer chemotherapeutic agents [36], the development of resistance interferes with the effectiveness of TMZ, especially in GMB. While O^6^-methylguanine-DNA methyltransferase (MGMT) repair activity and the uniquely resistant populations of glioma stem cells are relatively well-established contributing factors in the development of TMZ resistance, a number of other molecular mechanisms have been identified, including cytoprotective autophagy [37]. The potential influence of autophagy on resistance to TMZ has been studied largely in GMB, with only a few publications relating to melanoma. 

As indicated in some detail below, the bulk of studies in the literature have reported that TMZ induces autophagy in different tumor cell lines; however, the role of the induced autophagy has not been fully defined. 

Mechanistically, numerous molecular mechanisms have been proposed for the activation of autophagy by TMZ [37]. These include reactive oxygen species (ROS) accumulation, leading to activation of the MAPK/ERK signaling pathway and autophagy induction [38]. ATM/AMPK/ULK1 pathway activation, as well as inhibition of the PI3K/Akt/mTOR pathway, have been shown to mediate autophagy in response to TMZ [39,40]. TMZ-mediated autophagy can also occur under ER stress, where IRE1 activates XBP1, ASK1, and molecules downstream of JNK that promote autophagy [38,41] (Figure 1).

After absorption, Temozolomide undergoes hydrolysis to methyl triazeno imidazole carboxamide (MTIC). MTIC targets DNA, primarily alkylating guanine bases at the O6 position, as well as the N7 and N3 position of guanine and adenine bases, respectively. MTIC-mediated alkylation of DNA bases leads to DNA strand breakage, which ultimately causes cell death. Several molecular pathways have been reported whereby temozolomide triggers autophagy, including ROS/MAPK/ERK, ATM/AMPK/ULK1, and PI3K/Akt/mTOR signaling, as well as via ER stress, where IRE1 activates XBP1, ASK1, and molecules downstream of JNK.

In the sections below, we attempt to elucidate the functions of autophagy induced by TMZ and evaluate the possible utility of autophagy targeting as adjuvant therapy to increase the effectiveness of TMZ in the clinical setting. In the context of these evaluations, the utility and rigor of the experimental approaches, including the validity of autophagy related assays, are noted.

## 4. Temozolomide and Autophagy 

### 4.1. Glioblastoma

Katayama et al. [42] studied the ability of TMZ to promote autophagy-dependent generation of ATP, which can contribute to glioma cell survival. Temozolomide (TMZ) was shown to induce autophagy in the U251 glioma cell line, based on LC3-I to LC3-II conversion, as well as down-regulation of mTOR activity together with the mTOR downstream targets, S6K, and 4E-BP1. TMZ treatment was also shown to increase ATP levels in the U251 cells. Pharmacological autophagy inhibition using 3-MA suppressed the TMZ-mediated increase in ATP levels in a dose dependent manner, with an increasing percentage of cells undergoing multi-micronucleation, indicating that the inhibition of autophagy-induced ATP production increased non-apoptotic cell death associated with micronucleation. Suppression of autophagy by 3-MA was further confirmed by suppression of the TMZ-induced conversion of LC3-I to LC3-II. Moreover, genetic autophagy suppression by Beclin-1 knockdown mediated via shRNA resulted in a similar trend to the utilization of 3-MA, with suppression of ATP levels, an increased number of cells undergoing multinucleation and, most importantly, increased TMZ-mediated cytotoxicity. While Beclin-1 knockdown also resulted in an expected reduction in TMZ-induced conversion of LC3-I to LC3-II, TMZ treatment also resulted in the (inconsistent) appearance of autophagic cells in Beclin-shRNA expressing cells, highlighting the need for studies of additional autophagic markers, such as the degradation of SQSTM1/p62. These results suggested that TMZ mediated autophagy promoted an increase in ATP levels that protected the cells from the drug-induced cell death involving multi-micronucleation [42]. Similar results were generated utilizing the p53-mutant glioma cell lines, U373 and SF188, indicating the cytoprotective role for TMZ-mediated autophagy. 

Knizhnik et al. [43] investigated the influence of autophagy on the response to TMZ using LN-229 glioblastoma and U87-MG astrocytoma cell lines. TMZ induced autophagy in both cell lines, based on the generation of monodansylcadaverine (MDC) stained vacuoles. GFP-LC3 levels were also increased in GFP-LC3 transfected cells and LC3B-II levels were shown to be elevated via Western blotting in both LN-229 and U87-MG cells [43]. Importantly, pharmacologic autophagy inhibition with 3-MA in combination with TMZ promoted a significant increase in apoptosis along with the appearance of a marked necrosis, indicating a cytoprotective role of autophagy in both LN-229 and U87-MG cell lines. With regard to the direct mechanism of action of TMZ, TMZ-induced autophagy (MDC staining) was shown to be prevented by transfection of both LN-229 and U87-MG cells with the O^6^-methylguanine-DNA methyltransferase (MGMT) repair enzyme. These outcomes were further confirmed using the MGMT inhibitor, O^6^-benzylguanine (O^6^BG), in the MGMT-transfected LN-229 cells. Specifically, TMZ treatment increased the number of MDC stained vacuoles upon MGMT inhibition, indicating an inverse relationship between autophagy induction and MGMT expression. This inverse relationship was further validated by Cyto-ID staining, which serves as a selective marker of autolysosomes and early autophagic compartments [43]. Here, TMZ treatment induced an increase in Cyto-ID fluorescence in both cell lines, an effect completely abolished by MGMT expression, indicating that O^6^-methylguanine lesions induced by TMZ are required for autophagy induction.

Multiple studies showed that the DNA mismatch repair I (MMR) system is required for O^6^-methylguanine mediated apoptosis through the conversion of O^6^-methylguanine/T mispairs into secondary lesions [44,45,46]. In further mechanistic studies, these investigators [43] addressed the needed role of the DNA mismatch repair I (MMR) system in autophagy induction mediated by TMZ. Specifically, it was demonstrated that siRNA-mediated knockdown of MSH6, a component of the MSH2–MSH6 complex of the MMR repair system that recognizes and binds to O^6^MeG/T mismatches, completely inhibited TMZ mediated autophagy (as shown by decreased MDC positive stained cells) [43]. MMR dependent autophagy was further validated in another MSH6 deficient cell line, DLD-1colorectal adenocarcinoma cells, where no autophagy induction was observed upon TMZ treatment using MDC staining; conversely, re-expression of MSH6 restored the ability of DLD-1 cells to undergo autophagy after TMZ treatment. In addition, shRNA mediated down-regulation of RAD5, a key element of homologous recombination (HR), which is the major pathway for repairing double strand breaks (DSBs) in response to TMZ [35,47], significantly increased TMZ-mediated autophagy based upon MDC staining [43], suggesting that HR protects against autophagy (i.e., the DNA lesions are required for autophagy induction). 

The role of telangiectasia mutated protein (ATM) protein kinase, which is recruited to DSBs via the Mre11-RAD50-NBS1 (MRN) complex and activates signal transduction pathways essential for the regulation of cell cycle progression with DNA repair, was also investigated here; TMZ treatment resulted in ATM phosphorylation, which is the result of O^6^-methylguanine processing [48,49], together with an increase in LC3B-II. It was further reported that siRNA-mediated ATM downregulation suppressed the induction of autophagy (again using MDC staining), and triggered apoptosis. These studies again support the cytoprotective role of TMZ-mediated autophagy, as well as indicating that autophagy induction requires MMR and ATM, and is reduced by HR [43]. 

These authors also investigated the relationship between TMZ-mediated autophagy and senescence [43], which has demonstrated both associations and dissociations in different experimental model systems [1,50,51,52]. TMZ was shown to induce senescence in both the LN-229 and U87-MG cell lines, but to a somewhat lower extent in the U87-MG cells, as confirmed by β-galactosidase activity, the C_12_-FDG assay, X-gal staining, and the appearance of senescence-associated heterochromatic foci (SAHF). Here it should be noted that autophagy and senescence appear to virtually always appear together [1,43,53]. Importantly, autophagy inhibition via 3-MA completely abolished senescence after TMZ treatment, indicating that TMZ induced a cytoprotective form of autophagy that appears to trigger senescence and protect against apoptosis [43].

Consequently, Knizhnik et al. [43] proposed that O^6^-methylguanine is converted through replication and MMR into DSBs, which in turn induce both autophagy and senescence that are ATM-dependent, antagonizing DSBs-mediated apoptosis. In the same context, HR, the primary pathway for repair of O^6^-methylguanine-induced DSBs, protects against autophagy, senescence, and apoptosis.

Recently, Shi et al. [54] studied TMZ in combination with nicardipine, a dihydropyridine calcium channel antagonist [54] that demonstrated promising results in preclinical cancer models [55], using glioma stem cells (GSCs), including surgical specimen derived SU4 and SU5 cell lines. These GSCs proved to be highly resistant to TMZ, requiring drug concentrations higher than 400 μm to detect reduced cell viability using the CCK-8 assay. The viability of both cell lines was significantly reduced upon combining TMZ with nicardipine, along with an increase in the apoptotic population together with mitochondrial Bax accumulation. Interestingly, TMZ in combination with nicardipine promoted induction of p-mTOR, up regulation of p62/SQSTM1 protein levels, and increased expression of LC3, indicative of autophagy suppression. The mCherry/GFP assay [56] also revealed a higher yellow fluorescence in GSCs treated with TMZ combined with nicardipine than each drug alone, indicating impaired autolysosomes fusion. Furthermore, the mTOR inhibitory drug and autophagy inducer, rapamycin, reversed the effectiveness of the combination of TMZ with nicardipine with a reduction in apoptosis and reducing the ratio of the proapoptotic Bax protein to the anti-apoptotic Bcl-2 protein. The influence of nicardipine on TMZ sensitivity was further supported by studies in vivo in an orthotopic GSCs model; here, a longer median survival was evident for the combination group treatment compared to each drug alone. Taken together, these studies again support a cytoprotective role of autophagy in these cell lines, with the caveat that the sensitization by the nicardipine could, in theory, be derived from effects that are not limited to the modulation of autophagy. 

Ando et al. [57] studied the combination of the mitochondrial complex I inhibitor, JCI-20679, and TMZ in various glioblastoma cell lines. JCI-20679 enhanced TMZ-mediated anti-proliferative effects in the murine primary glioblastoma cells, U251, T98, A172, and the U87-MG human glioblastoma cell lines. Interestingly, they reported that JCI-20679 reduced the expression levels of LC3-II, indicative of autophagy inhibition, but also reduced p62/SQSTM1 levels, which would indicate promotion of basal autophagy. The JCI-20679 enhanced effect in combination with TMZ was also investigated in vivo, where the systemic administration of JCI-20679 and TMZ significantly inhibited the growth of U87-MG cells inoculated in mice [57], indicating the cytoprotective role of autophagy. However, as was the case with nicardipine, it cannot be certain that the influence of the JCI-20679 compound was exclusively through autophagy inhibition, given that the effects on LC3-II and p62/SQSTM1 appear to be contradictory.

The endoplasmic reticulum (ER) is a major compartment for secretory protein folding [58,59]. ER stress occurs when the capacity of the ER to fold proteins becomes saturated [60], leading to the accumulation of inactive or chemically aggressive proteins [61]. ER stress may be caused by factors that impair protein glycosylation, disulfide bond formation, or disturb (mutation or overexpression) proteins entering the secretory pathway [59]. ER stress causes the activation of two protein degradation pathways, the ubiquitin-proteasome via ER-assisted degradation, and lysosome-mediated protein degradation via autophagy [62]. The unfolded protein response (UPR) is a complex signal transduction pathway that is triggered by the activation of at least three UPR stress sensors: inositol-requiring protein 1 (IRE1), protein kinase RNA-like ER kinase (PERK), and activating transcription factor 6 (ATF6). These sensors affect almost every aspect of the secretory pathway, including protein folding, ER-associated degradation (ERAD), ER biogenesis, protein entry to the ER, secretion, and autophagy through both transcriptional and non-transcriptional responses [63]. Under normal physiological conditions, these stressors are inactivated by chaperone 78 kDa glucose-regulated protein (GRP78) [63]. GRP78 maintains ER integrity and assists in autophagosome formation independent of Beclin 1-dependent autophagy. GRP78 knockdown causes the suppression of autophagy induced by ER stress [64]. However, GRP78 knockdown did not generate promising results in the studies reported by Golden et al. [65] where siRNA mediated knockdown of GRP78 in the U251 cell line (p53 mutant, phosphatase and tensin homolog [PTEN] mutant) enhanced the inhibition of colony formation mediated by TMZ by a mere 10 to 20%. 

Golden et al. [65] also studied TMZ and CQ in various glioblastoma cell lines. Using a clonogenic survival assay, they showed that treating U251 and LN229 (p53 mutant, PTEN wild type) glioma cell lines with a combination of TMZ and CQ resulted in a more pronounced reduction in colony formation than each drug alone. 3-MA also enhanced the cytotoxic activity of TMZ; however, Beclin 1 knockdown via siRNA did not affect TMZ-mediated cytotoxicity, raising some concerns as to the cytoprotective function of autophagy in these experiments. 

Additional studies investigated CQ effects in TMZ-resistant (TMZ^R^) and their parental TMZ-sensitive (TMZ^S^) glioma cell lines [65]. Using the MTT assay (which is generally not considered to be the most rigorous or sensitive approach), CQ alone showed a significant toxicity in TMZ-resistant cell lines (U251-TMZ^R^, LN229-TMZ^R^, U87-TMZ^R^, and TuBECs) as compared to their sensitive counterparts (U251-TMZ^S^, LN229-TMZ^S^, U87-TMZ^S^, and BECs) [65]. CQ alone also demonstrated toxicity to both TMZ-sensitive and TMZ-resistant U87 glioma cell lines (p53 wild type, PTEN mutant). Using a clonogenic survival assay with TMZ^R^ and TMZ^S^ cells, the combination of CQ and TMZ resulted in higher efficacy than each drug alone. Importantly, whereas TMZ alone did not cause LC3-II accumulation, the combination of TMZ with CQ resulted in a significant increase in the levels of LC3B-II, ubiquitinated proteins, and cleaved PARP, as well as the ER stress pro-apoptotic protein CCAAT enhancer-binding protein (C-EBP) homologous protein (CHOP/GADD-153) in U251 glioma cells [65]. In vivo, nude mice implanted subcutaneously with U87 glioma cells treated with CQ in combination with TMZ displayed higher levels of CHOP/GADD-153 expression than the tumors treated with CQ or TMZ alone [65]. Although, a cytoprotective role of autophagy is suggested by these findings, additional genetic silencing studies and more autophagy markers would need to be measured [66], as conclusions cannot be based on LC3-II levels and TMZ in combination solely with CQ. As stated in [66], “For analysis of genetic inhibition, a minimum of two ATG genes (including for example BECN1, ATG7, LC3/GABARAP or ULK1) should be targeted to help ensure the phenotype is due to inhibition of autophagy”.

In contrast to the extensive evidence for a cytoprotective function of temozolomide-induced autophagy, the results by Kanzawa et al. [67], Lee et al. [68], and Torres et al. [69] suggested a cytotoxic function of autophagy in glioma cells.

Kanzawa et al. [67] examined the association of autophagy with TMZ using the U373-MG cell line. TMZ-induced autophagy in the U373-MG cell line was indicated by an increase in the number of autophagosomes and secondary lysosomes as assessed by electron microscopy, increased bright red fluorescence staining using acridine orange, and confirmed by the fluorescence intensity indicative of an increased number of punctate GFP-LC3 vacuoles. TMZ treatment also increased LC3 mRNA expression levels, although this is not necessarily considered a direct indication of the promotion of autophagy. 

In further experiments, TMZ-induced autophagy was pharmacologically inhibited using 3-MA or bafilomycin A1 (BAF A1). In general, inhibition of the early stages of (protective) autophagy with 3-MA, and the later stages of autophagy with bafilomycin will generate similar outcomes, specifically, enhanced drug sensitivity via the promotion of apoptosis [3,19,70]. Combining TMZ with bafilomycin A1 resulted in decreased cell viability, and increases in the apoptotic population and the activation of the apoptosis executioner caspase, caspase-3, as compared to each drug alone [67]. The combination of TMZ with BAF A1 induced loss of mitochondrial membrane potential as well as causing cathepsin B release from lysosomes, indicative of lysosomal membrane permeabilization [67]. These observations are all quite consistent with the autophagy induced by temozolomide being cytoprotective in function and the possibility that autophagy inhibition could enhance the therapeutic response. However, the outcome was largely the opposite (i.e., increased viability) when the cells were exposed to TMZ in combination with 3-MA. In addition, autophagy blockade with 3-MA interfered with sensitization to TMZ by bafilomycin A1. Furthermore, the TMZ-induced punctate pattern with GFP-LC3 was suppressed by 3-MA, consistent with 3-MA inhibiting autophagy prior to autophagosome membrane association of LC3 [67]. In this context, 3-MA inhibits LC3 incorporation into the membrane of autophagosomes, whereas bafilomycin A1 impairs autophagosome/lysosomes fusion [67]. These results are interesting in suggesting that early autophagy inhibition (3-MA) versus late autophagy inhibition (bafilomycin) can result in different outcomes in terms of sensitivity to TMZ. A possible explanation for these observations is that autophagic structures/autophagic vacuoles are being accumulated upon combining TMZ with bafilomycin A1, leading to cell cytotoxicity, as autophagosomal membranes could serve as a platform for an intracellular death-inducing signaling complex [71], and this autophagy-related cytotoxicity is abolished by a block of the early autophagic steps with 3-MA. Since clinical trials involving autophagy inhibition utilize the late-stage autophagy inhibitors chloroquine or hydroxychloroquine, the effects of bafilomycin A1 suggest that such a strategy might prove to be therapeutically effective. However, one limitation to these studies in terms of identifying the functional form of autophagy induced by TMZ is the absence of genetic inhibition experiments [66] to clearly define the role that autophagy plays in this cell line, i.e., to distinguish between the cytotoxic and cytoprotective functions.

Lee et al. [68] studied the impact of combining TMZ with CQ using U87-MG (wild type p53) and U373 (mutant p53) glioma cell lines. TMZ-induced autophagy was indicated by the generation of GFP-LC3 puncta. TMZ in combination with CQ markedly inhibited the proliferative ability of U87-MG cells compared to each drug alone with a significant increase in the apoptotic population as measured by PI staining and caspase activity. Bafilomycin also showed a similar trend as CQ when combined with TMZ, with a significant increase in inhibitory effects and a marked apoptosis in U87-MG cells. However, these synergistic effects between CQ and TMZ were nearly abolished upon knockdown of Beclin 1 mediated by siRNA or pretreatment with 3-MA. As with the studies by Kanzawa et al. [67], these results suggested that the accumulation of autophagic structures/autophagic vacuoles upon co-treatment with CQ and TMZ is cytotoxic, and this autophagy-related cytotoxicity is abolished by a block of the early autophagic steps with Beclin 1 siRNA or 3-MA. Interestingly, this work implicated p53 in drug action as p53 knockdown via siRNA severely abrogated the synergistic effects of combining CQ with TMZ. The combination effects were also suppressed in p53-mutant overexpressing U87 cells and in the p53 mutant U373 cell line, where no synergistic effects or enhanced apoptosis were observed for CQ in combination with TMZ. However, the involvement of p53 in modulation of TMZ sensitivity via autophagy inhibition are contradicted in a study by Katayama et al. [42], which suggest the lack of influence of p53 function on these responses. 

Torres et al. [69] investigated the effect of combining temozolomide with cannabinoids which suggest a cytotoxic function of autophagy. Screening the effect of Δ^9^-tetrahydrocannabinol (THC) in combination with temozolomide in vitro demonstrated a synergistic interaction in U87MG, LN405, HG14, and HG19 cells. However, here the THC but not the TMZ was shown to induce autophagy in U87-MG cells based on LC3-I or LC3-II levels. The combination of TMZ with THC was associated with enhanced accumulation of LC3-II as compared to each drug alone, with significant promotion of apoptosis, as shown by active caspase-3 immunostaining and confirmed using the pan-caspase inhibitor QVDOPH [72]. Pharmacological inhibition of autophagy using 3-MA, or genetically using siRNA directed against Atg1, prevented the THC + TMZ-induced cell death.

Further in vivo studies of TMZ combined with THC using tumor xenografts of U87-MG cells in immunodeficient mice demonstrated a significant reduction in tumor growth as compared to each drug alone. The combination of TMZ + THC significantly enhanced autophagy, as shown by LC3 immunostaining, and apoptosis as determined by TUNEL assay in these tumors as compared to each drug alone.

As mentioned previously, MGMT overexpression is one of the mechanisms that has been widely associated with resistance to TMZ [73]. Torres et al. [69] observed that T98G cell lines have significant higher MGMT mRNA levels than U87-MG cells, consistent with T98G cells demonstrating reduced sensitivity to TMZ. Tumor xenografts using T98G cells were markedly less sensitive to TMZ or THC alone as compared to their U87-MG counterparts, while treatment with TMZ + THC significantly reduced the growth of T98G-based tumors with a significant induction of both autophagy and apoptosis. 

Several publications have shown that cannabidiol (CBD) can reduce the growth of glioma xenografts [74,75]. Torres et al. [69] demonstrated that THC in combination with CBD greatly reduced the viability of U87-MG, HG19, T98G, HG2, HG21, U373, A172, SW1783, and LN405 glioma cells. These results were confirmed in vivo using U87-MG cell–derived subcutaneous xenografts. Furthermore, co-treatment with THC and CBD stimulated autophagy and apoptosis in vitro and in vivo as compared to each drug alone, while pharmacologic or genetic inhibition of autophagy inhibited THC + CBD mediated cell death.

Finally, the potential utility of THC + CBD combined with TMZ was investigated. The triple combination reduced the viability of U87-MG and T98G glioma cell lines with enhanced autophagy and apoptosis. Furthermore, pharmacological inhibition of autophagy with 3-MA prevented TMZ + THC + CBD-induced cell death. The effect of the triple combination was further validated in U87-MG cell-derived tumor xenografts with a reduction in tumor growth together with a significant induction of apoptosis and autophagy. These results strongly support a cytotoxic role of autophagy. However, it is curious that in this work, TMZ of itself was not shown to induce significant autophagy. 

With the possible exception of the studies involving the cannabinoids, the literature strongly leans towards a cytoprotective role of TMZ-induced autophagy in glioblastoma.

#### Clinical Trials 

A Phase I/II clinical trial conducted by Rosenfeld et al. [76] investigated the use of hydroxychloroquine (HCQ) together with radiation therapy and concurrent and adjuvant TMZ in patients diagnosed with glioblastoma multiforme. Patients received HCQ orally (200 mg to 800 mg daily) with radiation and concurrent and adjuvant TMZ; however, a dose of 600 mg/day HCQ did not consistently achieve autophagy inhibition, as shown by electron microscopy and immunoblotting assays; furthermore, no significant improvement in overall survival was reported. Sixteen phase I patients were evaluable for detection of dose-limiting toxicities; a dose of HCQ 800 mg/day resulted in significant side effects in a number of patients, including grade 3 and 4 neutropenia, thrombocytopenia, and sepsis. 

Recently, a phase IB trial conducted by Compter et al. [77] evaluated the potential use of CQ in combination with concurrent radiotherapy and temozolomide in patients diagnosed with glioblastoma. Patients received CQ orally (in the range between 200–400 mg) each daily for one week before starting the combination of temozolomide (75 mg/m^2^/day) with radiotherapy. Several adverse events were recorded including QTc prolongation, irreversible blurred vision, with the most common side effect being nausea/vomiting. These results reflect the serious side effects of the available autophagy inhibitors, CQ and HCQ, highlighting the need for clinically tolerable inhibitors, as well as providing the foundation for determining whether autophagy targeting can be considered an effective strategy for increasing the effectiveness of glioblastoma therapies. In Table 1, we summarize the completed and ongoing clinical trials that investigate the relation between TMZ and autophagy. 

### 4.2. Melanoma

The relationship between autophagy and temozolomide in melanoma has also been explored in a limited number of publications. However, the results do not appear to be as clear-cut as the findings in glioblastoma. Makita et al. [79] studied the effect of combining temozolomide with interferon (IFN-β) using different melanoma cell lines, including A375 and CRL-1579 cells. The combination of temozolomide with IFN-β showed a greater growth inhibitory response than temozolomide alone, as well as a significant increase in the apoptotic population, as shown by annexin V and propidium iodide (PI) assays. Importantly, they examined whether TMZ alone or in combination with IFN-β triggered autophagy in the melanoma cell lines. LC3 protein and Atg5/Atg12 complex protein expression levels were elevated following TMZ treatment in A375 and CRL-1579 cells, suggesting autophagy induction. Moreover, these protein levels were clearly increased after treatment with TMZ and IFN-β in A375 and CRL-1579 cells. Although suggestive of a cytotoxic role for TMZ-induced autophagy, such a conclusion would require more rigorous studies with both pharmacological and genetic autophagy inhibition [66].

Ryabaya et al. [80] studied the utilization of autophagy inhibitors, including chloroquine and LY294002 (LY) in combination with temozolomide in Mel MTP, Mel Z, Mel IL, Mel Ksen, and Mel Rac melanoma cells. TMZ in combination with CQ resulted in a quite modest 10–15% increases in anti-proliferative effect compared with TMZ alone. LY produced a greater reduction in cell proliferation in combination with TMZ (up to 30%) compared to TMZ alone except for the Mel IL cell line, which showed only a 15% reduction. These investigators further confirmed that the enhanced anti-proliferation effect mediated by CQ and LY are not related to *BRAF*-activating mutations (*BRAF*-activating mutations occur in 50–70% of melanoma cases) as shown by real-time PCR analysis. Using the annexin V/PI assay in Mel MTP, Mel Z, and Mel IL cell lines, CQ was shown to increase the extent of apoptosis by 15–20% as compared to TMZ alone; however, no significant apoptosis was reported with LY except with Mel MTP cells, which showed a two-fold increase in apoptosis (24% vs. 39.3%). While these results suggest that autophagy may play a modest cytoprotective role in these cell lines, the extent of sensitization observed when autophagy was inhibited is unlikely to prove to be of therapeutic benefit, if these preclinical results could be extrapolated to the clinic. 

Allavena et al. [81] studied trehalose, a natural disaccharide of glucose that has been identified as an mTOR-independent autophagy inducer [82] in the A375 and SK-Mel-28 melanoma cell lines. Trehalose in combination with TMZ did not confer additional anti-proliferative activity over TMZ alone in the A375 cells, results that were confirmed by measuring caspase-3 and -7 activity. However, in the long term clonogenic survival assay, trehalose significantly reduced colony formation ability in A375 cells, with a higher sensitivity to trehalose in combination with TMZ than to TMZ alone. Interestingly, TMZ alone did not induce significant autophagy, based on assessment of LC3-II and p62/SQSTM1 levels. Whereas SK-Mel-28 cells showed similar results to A375 cells with regard to autophagy induction by either TMZ or trehalose alone, trehalose combined with TMZ produced a significantly greater reduction in cell proliferation than TMZ alone. Trehalose in combination with TMZ also significantly reduced the colony formation ability of these melanoma cells compared to TMZ alone, an outcome that was further enhanced upon combination with radiation. These studies suggest that the promotion of autophagy (though not by TMZ) can sensitize the cells to TMZ; however, any conclusions are incomplete in the absence of studies involving pharmacological and genetic inhibition [66]. 

#### Clinical Trials

One Phase I clinical trial [78] in melanoma was conducted in 2014 where a promising result has been reported (see Table 1). HCQ in the range between 200 to 1200 mg was given orally on a daily basis to 40 patients (73% metastatic melanoma) in combination with dose intense oral TMZ 150 mg/m^2^ daily for 7 or 14 days. Autophagy inhibition was reported in response to the combined therapy with a significant accumulation of autophagic vacuoles in peripheral blood mononuclear cells. A partial responses and stable disease was observed in 3/22 (14%) and 6/22 (27%) patients with metastatic melanoma, respectively. In the final dose cohort, 2/6 patients with refractory *BRAF* wild-type melanoma had a close to complete response, and prolonged stable disease. This combination was well tolerated with no obvious recurrent dose-limiting toxicities. 

## 5. Conclusions

As summarized in Table 2, it remains uncertain whether TMZ is able to induce autophagy in melanoma cells or whether the targeting of autophagy could generate clinically relevant and positive outcomes. In contrast to the preclinical data, one Phase I clinical trial [78] has reported promising results; however, no data is available for autophagy inhibition in response to TMZ in clinical trials in melanoma since 2014. 

Although the results of preclinical studies in GMB are quite strongly suggestive of cytoprotective autophagy occurring in response to TMZ, we could not exclude the possibility of inducing autophagy as a therapeutic target. Nevertheless, proposing autophagy inhibition as a clinical strategy for the treatment of GMB in combination with temozolomide is fraught with serious limitations. These include the likely difficulty of achieving autophagy inhibition in the tumors due to the limited penetration of CQ [83] and HCQ [84,85,86] across the blood brain barrier. Furthermore, serious neurological adverse effects of the clinically available autophagy inhibitor HCQ have been reported in the literature; these include psychomotor agitation, irritability, nervousness, emotional changes [87], anxiety, and psychiatric symptoms [86]. Other potential side effects include respiratory failure, prolonged QT interval, and cardiomyopathy [88], emphasizing the need for more selective autophagy inhibitors (pre-clinical efforts [89,90]) with more favorable side effect profiles. 

Finally, another approach that is currently being considered is the relationship between glioma-initiating cells and autophagy, specifically whether autophagy inhibition [91] or induction [92] could be effective when the glioma initiating cells are considered as the primary drug target. 

## Figures and Tables

**Figure 1 cells-12-00535-f001:**
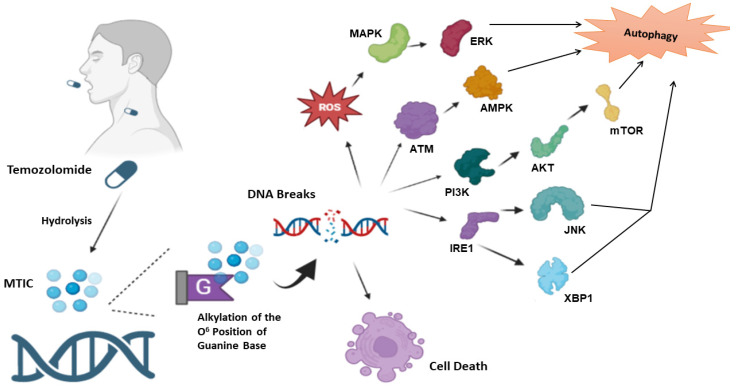
Mechanism of action of temozolomide and relationship with autophagy.

**Table 1 cells-12-00535-t001:** Clinical trials that investigate the relation between temozolomide and autophagy.

Title	Reference
Hydroxychloroquine, Radiation, and Temozolomide Treating Patients with Newly Diagnosed Glioblastoma Multiforme.	[76] [NCT00486603]
Chloroquine combined with concurrent radiotherapy and temozolomide for newly diagnosed glioblastoma: a phase IB trial.	[77]
A Phase II Randomized Controlled Trial for the Addition of Chloroquine, an Autophagy Inhibitor, to Concurrent Chemoradiation for Newly Diagnosed Glioblastoma.	[NCT02432417]
TN-TC11G (THC+CBD) Combination with Temozolomide and Radiotherapy in Patients With Newly-diagnosed Glioblastoma (GEINOCANN).	[NCT03529448]
The Addition of Chloroquine to Chemoradiation for Glioblastoma (CHLOROBRAIN).	[NCT02378532]
The Addition of Chloroquine to Chemoradiation for Glioblastoma,	[NCT02432417]
Phase I trial of hydroxychloroquine with dose-intense temozolomide in patients with advanced solid tumors and melanoma.	[78]

**Table 2 cells-12-00535-t002:** The roles of autophagy mediated by temozolomide in different cancer models.

Drug	Cell Line	Autophagy Modulation	Autophagy Function	References
Temozolomide	U251 glioma cell line and p53-mutant glioma cell lines, U373 and SF188.	3-MA, and Beclin-1 knockdown mediated via shRNA	Cytoprotective	[42]
	LN-229 glioblastoma and U87-MG astrocytoma cell lines.	3-MA	Cytoprotective	[43]
Temozolomide and nicardipine	Glioma stem cells (GSCs) including surgical specimen derived SU4 and SU5 cell lines.	Rapamycin	Cytoprotective	[54]
Temozolomide and JCI-20679	Murine primary glioblastoma cells, U251, T98, A172 and U87MG human glioblastoma cell lines.	Bafilomycin A1	Cytoprotective	[57]
Temozolomide	Resistant cell lines (U251-TMZ^R^, LN229-TMZ^R^, U87-TMZ^R^, and TuBECs) and their sensitive counterparts (U251-TMZ^S^, LN229-TMZ^S^, U87-TMZ^S^, and BECs). In vivo.	CQ, 3-MA, and knockdown of Beclin 1 mediated by siRNA	Cytoprotective	[65]
Temozolomide	U373-MG cell line	3-Methyladenine (3-MA), and bafilomycin A1	Cytotoxic	[67]
Temozolomide	U87-MG (wild type p53), U373 (mutant p53) glioma cell lines, and P53-overexpressing U87 mutant cell line.	CQ, bafilomycin A1, 3-MA, and knockdown of Beclin 1 mediated by siRNA	Cytotoxic in U87-MG cells Dependent upon p53 status, non-protective in U373 (mutant p53) glioma cell lines and p53-overexpressing U87 mutant cell line	[68]
Temozolomide, Δ^9^-tetrahydrocannabinol (THC) and Cannabidiol (CBD)	U87MG, LN405, HG14, HG19, T98G, HG2, HG21, U373, A172, SW1783 cells, T98G and U87-MG based tumor xenografts.	3-MA, or genetically using siRNA directed to Atg1	Cytotoxic form of autophagy TMZ did not induce autophagy	[69]
Temozolomide and interferon (IFN-β)	Melanoma cell lines including A375 and CRL-1579 cells	NA	Cytotoxic	[79]
Temozolomide	Mel MTP, Mel Z, Mel IL, Mel Ksen, and Mel Rac melanoma cell lines.	Chloroquine and LY294002 (LY)	Cytoprotective	[80]
Temozolomide and trehalose with and without radiation	A375 and SK-Mel-28 melanoma cells	NA	TMZ did not induce autophagy	[81]

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
