# Peer review of "Is Autophagy Inhibition in Combination with Temozolomide a Therapeutically Viable Strategy?"

_cells, 2023, doi:10.3390/cells12040535_

Round 1
Reviewer 1 Report
1.It is suggested to add new in vitro and in vivo research studies about the eco-toxicological concerns.
2.what is the suggestion of this study for future works?
3.Please discuss and compare your results with previous works and add suggestions including Nano antioxidants based
methods.
4.It will be better to add the role on mitochondria.
5.Please add details for time period and dose selection from literatures.
6.More references for the discussion part of manuscript and update and bold your study novelty should be added: e.g.,
-DOI: 10.1186/s12885-022-09775-y
-DOI: 10.1016/j.biopha.2018.01.117
Author Response
Reviewer 1:
- It is suggested to add new in vitro and in vivo research studies about the eco-toxicological concerns.
A: Unfortunately, we were unable to understand what was meant by “ecotoxicological concerns”. Our current manuscript focuses on the role of autophagy in response to Temozolomide in glioblastoma and melanoma. We did not recognize the relationship of our work to ecotoxicological concerns.
- What is the suggestion of this study for future works?
A: We appreciate this comment. In our discussion, we emphasize on the necessity for the development of more specific autophagy inhibitors with more favorable side effect profiles than CQ and HCQ. We further highlight the importance of developing autophagy inhibitors that can cross the blood-brain barrier with regard to the treatment of glioblastoma.
- Please discuss and compare your results with previous works and add suggestions including Nano antioxidants based methods.
A: Thank you for this suggestion. We will consider including work on nanoparticle delivery systems and the possible utilization of antioxidants in future manuscripts. These concepts would not integrate well into the current work.
- It will be better to add the role on mitochondria.
A: Thank you for your comment. In fact, we had included reference to mitochondrial membrane potential change in response to temozolomide (Kanzawa, T., et al., Role of autophagy in temozolomide-induced cytotoxicity for malignant glioma cells. Cell Death & Differentiation, 2004. 11(4): p. 448-457), as well as the utilization of the mitochondrial complex I inhibitor, JCI-20679, and TMZ in glioma (Ando, S., et al., JCI-20679 suppresses autophagy and enhances temozolomide-mediated growth inhibition of glioblastoma cells. Biochem Biophys Res Commun, 2022. 591: p. 62-67) .
- Please add details for time period and dose selection from literatures.
A: Thank you for your comment. In fact, we made every effort to review the entire literature with regard to the role of autophagy in response to temozolomide in the malignancies where this drug is utilized.
- More references for the discussion part of manuscript and update and bold your study novelty should be added: e.g.,
-DOI: 10.1186/s12885-022-09775-y
-DOI: 10.1016/j.biopha.2018.01.117
A: Thank you for your suggestion. Unfortunately, these citations do not fall within the scope of this article that focuses on the role of autophagy in response to Temozolomide. The first citation relates to the targeting of autophagy in response to proteasome inhibitors while the second citation discusses azilsartan and its role in hepatocellular carcinoma. However, we do plan to consider the first citation in a future manuscript investigating the role of autophagy in response to different chemotherapeutic modalities.
Reviewer 2 Report
In this review article, the authors explore the possibility of counteracting autophagy mechanisms to increase the effectiveness of Temozolomide based chemotherapy. Effectively, in cancer therapy combination of drugs is largely used in clinical practice, and the possibility to modulate autophagy has been described in many studies as an important way to increase the efficiency of therapies. This is due because autophagy is a master pathway that may be activated in response to stresses, as chemotherapy, and may limit the efficacy of therapy. For this reason, inhibiting autophagy, that is activated by TMZ , should be an interesting opportunity to increase chemotherapy. The review is overall well-written, except the introduction that is surely too short, and in general it describes different aspects related to the concept that authors wish to communicate.
-
Minor style: line 30, there are () , please correct or include text within.
-
The introduction should be extended and should describe better the goals of the manuscript. For example, some sentences discussing previous publications of the authors should be reported, as well as a brief description of autophagy mechanisms, with a focus on the dual role of autophagy in cancer (pro-cell death or survival).
-
In section 2 “Autophagy Overview” the authors should mention the lysosomal system (with a focus on TFEB transcription factor since its great important in autophagy regulation) (doi: 10.1038/s41556-020-0549-1, doi: 10.3390/cells10102752) and its dysfunction in cancer. More details about cell death autophagy should be reported in this section (10.1038/s12276-020-0455-4). A brief/easy scheme reporting autophagy mechanism and autophagosome formation should be included.
-
More focus should be given to the mechanisms of autophagy modulation in cancer therapy as introductive aspect, and connect them with the section “4.1.1. Clinical trials”. Also, although this review aims to inhibit autophagy as a therapeutic way to increase effect of TMZ, the authors should be mention that, in other case, autophagy may also be induced for therapeutic purposes , for example by blocking mTOR/AKT pathways, and show some clinical aspects. Many papers/articles showing the potential of modulating autophagy are reported and listed here: (doi: 10.3390/curroncol29030141, doi: 10.3389/fphar.2020.01141)
-
In the section 4. “Temozolomide and Autophagy” the paragraph relative to glioblastoma is very dense with a lot of information and should be simplified. The authors commented on a number of papers in a very detailed manner (from ref 35 to ref 55) . I suggest reducing and simplifying this paragraph and organizing it for a mechanism of action induced by TMZ. A novel scheme may help authors.
-
What is the effect of blocking autophagy after TMZ treatment in Glioma Initiating Cells (GICs) ? Since the importance of GICs in glioma disease, the authors should mention their role and how they lead to chemotherapy failure; hence should be clearly expressed that targeting GICs is a promising cancer therapy adopted in different pre-clinical studies.
- In continuation with the point above reported, some important literature should be added and carefully discussed: specifically the possibility to combine TMZ with cannabinoids (a pro-inducing autophagy agents) could drastically improve the efficiency of chemotherapy through modulation of signaling pathways and autophagy 10.1016/j.bcp.2018.09.007. doi: 10.1158/1535-7163.MCT-10-0688This last point is a clear demonstration that TMZ may be enhanced also with inducing-autophagy agents and not exclusively with its inhibition.
Author Response
Reviewer 2 :
In this review article, the authors explore the possibility of counteracting autophagy mechanisms to increase the effectiveness of Temozolomide based chemotherapy. Effectively, in cancer therapy combination of drugs is largely used in clinical practice, and the possibility to modulate autophagy has been described in many studies as an important way to increase the efficiency of therapies. This is due because autophagy is a master pathway that may be activated in response to stresses, as chemotherapy, and may limit the efficacy of therapy. For this reason, inhibiting autophagy, that is activated by TMZ, should be an interesting opportunity to increase chemotherapy. The review is overall well-written, except the introduction that is surely too short, and in general it describes different aspects related to the concept that authors wish to communicate.
- Minor style: line 30, there are (), please correct or include text within.
A: Thank you for your guidance; we adjusted this error and included the necessary reference.
- The introduction should be extended and should describe better the goals of the manuscript. For example, some sentences discussing previous publications of the authors should be reported, as well as a brief description of autophagy mechanisms, with a focus on the dual role of autophagy in cancer (pro-cell death or survival).
A: Thank you for your guidance. In fact, our introduction refers to our previously published articles that discuss the role of autophagy in response to different therapeutic modalities. Regarding the mechanisms/functions of autophagy, this is discussed in detail in our previous review article as a part of this series, DOI: 10.3390/cancers14174289. The different roles of autophagy ( cytotoxic, cytoprotective, cytostatic and non-protective) are described in our introduction and in our previous publications, DOI: 10.3390/biomedicines10071632.
- In section 2 “Autophagy Overview” the authors should mention the lysosomal system (with a focus on TFEB transcription factor since its great important in autophagy regulation) (doi: 1038/s41556-020-0549-1, doi: 10.3390/cells10102752) and its dysfunction in cancer. More details about cell death autophagy should be reported in this section (10.1038/s12276-020-0455-4). A brief/easy scheme reporting autophagy mechanism and autophagosome formation should be included.
A: Thank you for your guidance. We discuss the mechanism of autophagy in details in our previous review article as a part of this series of papers that address the role of autophagy in response to different therapeutic modalities, DOI: 10.3390/cancers14174289, DOI: 10.3390/biomedicines10071632. We have now added a sentence that directs the readers to the previous article where the mechanism of autophagy is described as follow “For more details, the mechanism of autophagy discussed by Rubinsztein et al. [18] and in our previous publications and also [3, 4].”.
- More focus should be given to the mechanisms of autophagy modulation in cancer therapy as introductive aspect, and connect them with the section “4.1.1. Clinical trials”. Also, although this review aims to inhibit autophagy as a therapeutic way to increase effect of TMZ, the authors should be mention that, in other case, autophagy may also be induced for therapeutic purposes , for example by blocking mTOR/AKT pathways, and show some clinical aspects. Many papers/articles showing the potential of modulating autophagy are reported and listed here: (doi: 3390/curroncol29030141, doi: 10.3389/fphar.2020.01141)
A: Thank you for your guidance. As per the reviewer’s suggestion, we now mention the possibility of promoting autophagy for therapeutic purposes in our discussion as follow “Although, the results of preclinical studies in GMB are quite strongly suggestive of cytoprotective autophagy occurring in response to TMZ, we could not exclude the possibility of inducing autophagy as a therapeutic target”. Also, we refer in our introduction to the possible mechanisms for autophagy modulation with suitable references.
- In the section 4. “Temozolomide and Autophagy” the paragraph relative to glioblastoma is very dense with a lot of information and should be simplified. The authors commented on a number of papers in a very detailed manner (from ref 35 to ref 55) . I suggest reducing and simplifying this paragraph and organizing it for a mechanism of action induced by TMZ. A novel scheme may help authors.
A: Thank you for your guidance. In response to the reviewer’s suggestion, we have simplified this section by dividing it into sub-paragraphs that should be easier to follow. Details relating to the mechanism of action of TMZ are provided in section 3 with a figure.
- What is the effect of blocking autophagy after TMZ treatment in Glioma Initiating Cells (GICs) ? Since the importance of GICs in glioma disease, the authors should mention their role and how they lead to chemotherapy failure; hence should be clearly expressed that targeting GICs is a promising cancer therapy adopted in different pre-clinical studies.
A: Thank you for your guidance. We had overlooked the potential role of glioma initiating cells and now added this element to the Discussion with the relevant references.
- In continuation with the point above reported, some important literature should be added and carefully discussed: specifically, the possibility to combine TMZ with cannabinoids (a pro-inducing autophagy agents) could drastically improve the efficiency of chemotherapy through modulation of signaling pathways and autophagy 1016/j.bcp.2018.09.007. doi: 10.1158/1535-7163.MCT-10-0688. This last point is a clear demonstration that TMZ may be enhanced also with inducing-autophagy agents and not exclusively with its inhibition.
A: Thank you for this suggestion. We now provide a relatively detailed and comprehensive section relating to studies where TMZ is combined with cannabinoids and autophagy appears to express a cytotoxic function.
Reviewer 3 Report
The review is well organized and very well written. The content will be of interest to pre-clinical and clinical researchers in the autophagy-oncology field. Particularly appreciated is the scientifically sound, critical and objective style - and the important inclusion of the descriptions of the autophagy methods that have been used in the various papers.
Although the overall scientific conclusions are sound and the review is valuable, there are a few important shortcomings in the manuscript that should be carefully addressed before the paper can be suitable for publication.
Major points:
1) The authors wrongly conclude that the paper by Lee et al. (Ref 43: The synergistic effect of combination temozolomide and chloroquine treatment is dependent on autophagy formation and p53 status in glioma cells. Cancer Lett, 2015. 360(2): p. 195-204) suggest a cytoprotective function of autophagy. The authors state on page 6, lines 252-254 that: “These synergistic effects between CQ and TMZ were nearly abolished upon knockdown of Beclin 1 mediated by siRNA or pretreatment with 3-MA, indicating that the synergy was dependent upon the cytoprotective function of autophagy.”. A more likely conclusion from the results of Lee et al. is that the accumulation of autophagic structures / autophagic vacuoles upon co-treatment with CQ and TMZ is cytotoxic, and this autophagy-related cytotoxicity is abolished by a block of the early autophagic steps with Beclin 1 siRNA or 3-MA.
2) The authors describe the data obtained with 3-MA and bafilomycin by Kanzawa et al (Ref 36: Role of autophagy in temozolomide-induced cytotoxicity for malignant glioma cells. Cell Death & Differentiation, 2004. 11(4): p. 448-457) as “inconsistent” (page 4, line 145) and “…unusual in suggesting that early autophagy inhibition (3-MA) versus late autophagy inhibition (Bafilomycin) can result in different outcomes in terms of sensitivity to TMZ.” (page 4, lines 163-165). However, there is actually a clear parallel to the findings of Lee et al. (ref 43) – as mentioned in point 1 above – which may indicate a cytotoxic role of accumulated autophagic structures / autophagic vacuoles when combining TMZ with either Baf or CQ. As one conceivable explanation to these findings of autophagy-related cytotoxic by Lee et al. and Kanzawa et al, it has previously been reported that autophagosomal membranes can serve as platforms for intracellular death-inducing signaling complexes (see e.g. M.M. Young, Y. Takahashi, O. Khan, S. Park, T. Hori, J. Yun, A.K. Sharma, S. Amin, C.D. Hu, J. Zhang, M. Kester, H.G. Wang, J Biol Chem 287 (2012) 12455-12468.). Other perfectly conceivable explanations also exist, so these observations are not necessarily "inconsistent" or "unusual".
3) According to pubpeer.com, the paper by Qu et al. (Ref 56. Berberine reduces temozolomide resistance by inducing autophagy via the ERK1/2 signaling pathway in glioblastoma. Cancer Cell Int, 2020. 20(1): p. 592) is associated with scientific misconduct, with the errors having been acknowledged by the corresponding author, who has filed a request to the Journal to retract the paper (see https://pubpeer.com/publications/30998A1D6100499DF6B429452359BA). There is evidence to suggest that data have been taken from other papers and presented as something else than what they are (in multiple figures). The authors should carefully evaluate this information and consider whether to include the findings from Qu et al. in their review or not (if including them, it should be explicitly explained in the point-by-point reply why the authors choose to do so).
Other points:
4) As a general point, the authors may wish to explicitly refer to one of the major points in the Guidelines for the use and interpretation of assays for monitoring autophagy (4th edition), on which the corresponding author is a co-author (Klionsky et al., Autophagy. 2021 Jan;17(1):1-382. doi: 10.1080/15548627.2020.1797280). As stated in “Quick Guide” point 4: “For analysis of genetic inhibition, a minimum of two ATG genes (including for example BECN1, ATG7, LC3/ GABARAP or ULK1) should be targeted to help ensure the phenotype is due to inhibition of autophagy.”. This is important to point out, since none of the reviewed papers have followed the Guidelines (only data with knockdown of BECN1 has been shown in some of the papers).
5) On page 4, line 162 the authors state that “…bafilomycin A1 impairs autophagosome/lysosomes fusion…”. This is a bit inaccurate, since the main effect of bafilomycin A1 is to de-acidify lysosomes and autolysosomes, whereas, on the other hand, the main autophagy-inhibitory effect of chloroquine seems to be to block autophagosome-lysosomes fusion (see e.g. Mauthe et al., Autophagy. 2018; 14(8):1435-55. doi: 10.1080/15548627.2018.1474314).
6) On page 6, lines 240-241 the authors state that “Here it should be noted that autophagy and senescence appear to virtually always appear together.” There are no references to this statement. Is the statement that of the paper in question (Ref 39. Knizhnik, A.V., et al., Survival and death strategies in glioma cells: autophagy, senescence and apoptosis triggered by a single type of temozolomide-induced DNA damage. PLoS One, 2013. 8(1): p. e55665.)? Or is it a statement by the authors? In that case, the authors should back up this claim with references to published papers.
7) Table 1 – information about drug and radiation dosages and key results/observations of the various clinical trials should be included in the Table. Page 8, lines 390-391: “In Table (1), we summarize most of the completed and ongoing clinical trials that investigate the relation between TMZ and autophagy.”. Minor point: It should say “Table 1”. (not “Table (1)”). Major point: why not show ALL completed and ongoing clinical trials that investigate the relation between TMZ and autophagy (instead of “most” of them)? Which trials have been omitted, and why?
8) Page 8, line 382: “Patients received CQ orally on a daily base one week…”. What was the dose of CQ?
9) The authors should go through the referencing and reference list carefully, since there are several mistakes, e.g.
- page 5, line 195 wrong reference name and number (it says “ Kanazawa [28]” but what is meant is “Kanzawa et al. [36]
- In Ref 5, the Journal name is missing: ELSHAZLY, A.-M., T.-V.-V. NGUYEN, and D.-A. GEWIRTZ, Is autophagy induction by PARP inhibitors a target for therapeutic benefit? 2022. 30(1): p. 1--12.
- Refs 30 and 31 are duplicates:
30. Singh, N., et al., Mechanisms of temozolomide resistance in glioblastoma - a comprehensive review. Cancer Drug Resist, 2021. 4(1): p. 526 17-43. 527
31. Singh, N., et al., Mechanisms of temozolomide resistance in glioblastoma - a comprehensive review. Cancer Drug Resistance, 2021. 4(1): 528 p. 17-43.
10) The manuscript should be carefully proofread, since there are some small mistakes here and there.
Author Response
Reviewer 3:
The review is well organized and very well written. The content will be of interest to pre-clinical and clinical researchers in the autophagy-oncology field. Particularly appreciated is the scientifically sound, critical and objective style - and the important inclusion of the descriptions of the autophagy methods that have been used in the various papers.
Although the overall scientific conclusions are sound and the review is valuable, there are a few important shortcomings in the manuscript that should be carefully addressed before the paper can be suitable for publication.
Major points:
- The authors wrongly conclude that the paper by Lee et al. (Ref 43: The synergistic effect of combination temozolomide and chloroquine treatment is dependent on autophagy formation and p53 status in glioma cells. Cancer Lett, 2015. 360(2): p. 195-204) suggest a cytoprotective function of autophagy. The authors state on page 6, lines 252-254 that: “These synergistic effects between CQ and TMZ were nearly abolished upon knockdown of Beclin 1 mediated by siRNA or pretreatment with 3-MA, indicating that the synergy was dependent upon the cytoprotective function of autophagy.”. A more likely conclusion from the results of Lee et al. is that the accumulation of autophagic structures / autophagic vacuoles upon co-treatment with CQ and TMZ is cytotoxic, and this autophagy-related cytotoxicity is abolished by a block of the early autophagic steps with Beclin 1 siRNA or 3-MA.
A: Thank you for this insightful suggestion. We have now modified this section to include the reviewer’s perspective on this paper by Lee.
- The authors describe the data obtained with 3-MA and bafilomycin by Kanzawa et al (Ref 36: Role of autophagy in temozolomide-induced cytotoxicity for malignant glioma cells. Cell Death & Differentiation, 2004. 11(4): p. 448-457) as “inconsistent” (page 4, line 145) and “…unusual in suggesting that early autophagy inhibition (3-MA) versus late autophagy inhibition (Bafilomycin) can result in different outcomes in terms of sensitivity to TMZ.” (page 4, lines 163-165). However, there is actually a clear parallel to the findings of Lee et al. (ref 43) – as mentioned in point 1 above – which may indicate a cytotoxic role of accumulated autophagic structures / autophagic vacuoles when combining TMZ with either Baf or CQ. As one conceivable explanation to these findings of autophagy-related cytotoxic by Lee et al. and Kanzawa et al, it has previously been reported that autophagosomal membranes can serve as platforms for intracellular death-inducing signaling complexes (see e.g. M.M. Young, Y. Takahashi, O. Khan, S. Park, T. Hori, J. Yun, A.K. Sharma, S. Amin, C.D. Hu, J. Zhang, M. Kester, H.G. Wang, J Biol Chem 287 (2012) 12455-12468.). Other perfectly conceivable explanations also exist, so these observations are not necessarily "inconsistent" or "unusual".
A: As above, the reviewer has provided a thoughtful and insightful analysis of the findings in the paper by Kanazawa et al, and we have modified this section of the manuscript to reflect the reviewer’s interpretation of the data.
3) According to pubpeer.com, the paper by Qu et al. (Ref 56. Berberine reduces temozolomide resistance by inducing autophagy via the ERK1/2 signaling pathway in glioblastoma. Cancer Cell Int, 2020. 20(1): p. 592) is associated with scientific misconduct, with the errors having been acknowledged by the corresponding author, who has filed a request to the Journal to retract the paper (see https://pubpeer.com/publications/30998A1D6100499DF6B429452359BA). There is evidence to suggest that data have been taken from other papers and presented as something else than what they are (in multiple figures). The authors should carefully evaluate this information and consider whether to include the findings from Qu et al. in their review or not (if including them, it should be explicitly explained in the point-by-point reply why the authors choose to do so).
A: Thank you for your guidance. As per the reviewer’s suggestion, we have re-evaluated this paper and replaced it with a different study, specifically doi: 10.1158/1535-7163.MCT-10-0688.
Other points:
4) As a general point, the authors may wish to explicitly refer to one of the major points in the Guidelines for the use and interpretation of assays for monitoring autophagy (4th edition), on which the corresponding author is a co-author (Klionsky et al., Autophagy. 2021 Jan;17(1):1-382. doi: 10.1080/15548627.2020.1797280). As stated in “Quick Guide” point 4: “For analysis of genetic inhibition, a minimum of two ATG genes (including for example BECN1, ATG7, LC3/ GABARAP or ULK1) should be targeted to help ensure the phenotype is due to inhibition of autophagy.”. This is important to point out, since none of the reviewed papers have followed the Guidelines (only data with knockdown of BECN1 has been shown in some of the papers).
Again, we appreciate the quite knowledgeable guidance from this reviewer. We have added this information from the Guidelines by Klionsky et al and acknowledge its importance since, as the reviewer notes, virtually none of the papers cited in our manuscript rigorously followed these guidelines.
5) On page 4, line 162 the authors state that “…bafilomycin A1 impairs autophagosome/lysosomes fusion…”. This is a bit inaccurate, since the main effect of bafilomycin A1 is to de-acidify lysosomes and autolysosomes, whereas, on the other hand, the main autophagy-inhibitory effect of chloroquine seems to be to block autophagosome-lysosomes fusion (see e.g. Mauthe et al., Autophagy. 2018; 14(8):1435-55. doi: 10.1080/15548627.2018.1474314).
A: Here we respectfully both agree and disagree with the (very knowledgeable) reviewer’s comment. We agree with the second part of the comment as CQ is an inhibitor for late stage of autophagy by blocking the fusion between autophagosomes and lysosomes (DOI: 10.1080/15548627.2018.1474314). Regarding BAF A1, a number of publication have shown that BAF disrupts Ca-P60A/SERCA-dependent autophagosome-lysosome fusion, as in (https://doi.org/10.1038/ncomms8007, doi:10.1080/15548627.2015.1066957, https://doi.org/10.1016/j.jmb.2019.10.028, https://doi.org/10.4161/auto.6845).
6) On page 6, lines 240-241 the authors state that “Here it should be noted that autophagy and senescence appear to virtually always appear together.” There are no references to this statement. Is the statement that of the paper in question (Ref 39. Knizhnik, A.V., et al., Survival and death strategies in glioma cells: autophagy, senescence and apoptosis triggered by a single type of temozolomide-induced DNA damage. PLoS One, 2013. 8(1): p. e55665.)? Or is it a statement by the authors? In that case, the authors should back up this claim with references to published papers.
A: Thank you for noting this omission. We have now included the omitted references.
7) Table 1 – information about drug and radiation dosages and key results/observations of the various clinical trials should be included in the Table. Page 8, lines 390-391: “In Table (1), we summarize most of the completed and ongoing clinical trials that investigate the relation between TMZ and autophagy.”. Minor point: It should say “Table 1”. (not “Table (1)”). Major point: why not show ALL completed and ongoing clinical trials that investigate the relation between TMZ and autophagy (instead of “most” of them)? Which trials have been omitted, and why?
A: Thank you for noting this error. Regarding the statement in clinical trials section, we adjusted it as follow, “we summarize the completed and ongoing clinical trials that investigate the relation between TMZ and autophagy.” We demonstrate the clinical trials that have a relation with our article scope (the autophagic role in response to temozolomide) in the main text but wanted to mention other autophagy related clinical trials as a source for the readers.
8) Page 8, line 382: “Patients received CQ orally on a daily base one week…”. What was the dose of CQ?
A: Thank you very catching this omission. We have now included the dose range (200-400 mg).
9) The authors should go through the referencing and reference list carefully, since there are several mistakes, e.g.
- page 5, line 195 wrong reference name and number (it says “ Kanazawa [28]” but what is meant is “Kanzawa et al. [36]
- In Ref 5, the Journal name is missing: ELSHAZLY, A.-M., T.-V.-V. NGUYEN, and D.-A. GEWIRTZ, Is autophagy induction by PARP inhibitors a target for therapeutic benefit? 2022. 30(1): p. 1--12.
- Refs 30 and 31 are duplicates:
- Singh, N., et al., Mechanisms of temozolomide resistance in glioblastoma - a comprehensive review. Cancer Drug Resist, 2021. 4(1): p. 526 17-43. 527
- Singh, N., et al., Mechanisms of temozolomide resistance in glioblastoma - a comprehensive review. Cancer Drug Resistance, 2021. 4(1): 528 p. 17-43.
A: Thank you for the extremely careful reading of our manuscript. We carefully rechecked the manuscript and corrected the errors and omissions.
10) The manuscript should be carefully proofread, since there are some small mistakes here and there.
A: Thank you for the extremely careful reading of our manuscript. We carefully rechecked the manuscript and corrected the errors and omissions.
Round 2
Reviewer 2 Report
The authors well responsed to all the comments. I believe the quality of the revised manuscript meets the standard for publication.
Reviewer 3 Report
The authors have satisfactorily addressed my concerns. I recommend the paper to be accepted.